# Identification and Evaluation of Diploid and Tetraploid *Passiflora edulis* Sims

**DOI:** 10.3390/plants13182603

**Published:** 2024-09-18

**Authors:** Xin Su, Xue Wang, Ruilian Li, Chiyu Zhou, Lin Chen, Shi Chen, Nianhui Cai, Yulan Xu

**Affiliations:** 1Key Laboratory of Forest Resources Conservation and Utilization in the Southwest Mountains of China, Ministry of Education, Southwest Forestry University, Kunming 650224, China; suxin@swfu.edu.cn (X.S.); wangwang199710@163.com (X.W.); fbfff@swfu.edu.cn (R.L.); zhouchiyu@swfu.edu.cn (C.Z.); cainianhui@swfu.edu.cn (N.C.); 2Key Laboratory of National Forestry and Grassland Administration on Biodiversity Conservation in Southwest China, Southwest Forestry University, Kunming 650224, China; linchen@swfu.edu.cn (L.C.); chenshi@swfu.edu.cn (S.C.)

**Keywords:** *Passiflora edulis* Sims, tetraploid, polyploidization, morphology traits, physiology and biochemistry, cold stress

## Abstract

*Passiflora edulis* Sims (2n = 18) is a perennial plant with high utilization values, but its spontaneous polyploidy in nature has yet to be seen. Thus, this study aims to enhance our understanding of polyploidy *P. edulis* and provide rudimentary knowledge for breeding new cultivars. In this study, colchicine-induced tetraploid *P. edulis* (2n = 36) was used as experimental material (T1, T2, and T3) to explore the variances between it and its diploid counterpart in morphology, physiology, and biochemical characteristics, and a comparison of their performance under cold stress was conducted. We measured and collected data on phenotype parameters, chlorophyll contents, chlorophyll fluorescence, photosynthesis, osmotic substances, and antioxidant enzymes. The results showed that tetraploid *P. edulis* exhibited a shorter phenotype, more giant leaves, darker leaf color, and longer and fewer roots. Moreover, the physiological and biochemical analysis indicated that the tetraploid *P. edulis* had better photosynthesis systems and higher chlorophyll fluorescence parameters than the diploid *P. edulis*. Additionally, the tetraploid *P. edulis* had higher activity of antioxidant enzymes (SOD, POD, CAT) and lower MDA content to maintain better resistance in low temperatures. Overall, we conclude that there were apparent differences in the morphological, physiological, and biochemical traits of the tetraploid and diploid *P. edulis*. The tetraploid plants showed better photosynthesis systems, higher osmotic substance content, and antioxidant enzyme activity than the diploid, even under cold stress. Our results suggest that tetraploids with more abundant phenotype variation and better physiological and biochemical traits may be used as a new genetic germplasm resource for producing new *P. edulis* cultivars.

## 1. Introduction

*Passiflora edulis* Sims is a perennial vine plant belonging to the Passifloraceae family. It is widely distributed in tropical and subtropical areas, boasting edible, medicinal, and ornamental values [1,2]. Its fruit is popular in the food processing industry and has broad market prospects owing to its rich nutrient elements and unique flavors [3]. Thus, cultivating commercial *P. edulis* varieties with high quality is of great importance.

Polyploidization refers to increases in genome DNA content, is a dominant feature of extant plant diversity [4], and is the main driving force in the evolution of plants [5]. After multiplying their chromosomes, different plant species’ morphological characteristics change to various degrees. From the perspective of most reports about polyploidy, it can be seen that larger stoma [6], thicker leaves [7], darker leaf color [8], increased flower diameter [9,10], stem diameter [11], and fruit sizes [12] presented in plants with higher ploidy. A higher organic nutrient content and better environmental stress resistance were observed in polyploids regarding physiological and biochemical traits. For example, tetraploid *Eucalyptus urophylla* reported higher net photosynthetic rates and a higher content of specific secondary metabolites [13], and the tetraploid line of *Centella asiatica* (L.) exhibited higher total triterpenoids than its diploid counterpart [14]. In addition, tetraploid bermudagrass “CX” showed significantly greater shade tolerance compared to its triploid bermudagrass, for the former can maintain higher photosynthetic performance, photosynthetic pigments, and a lower efficient use of light energy under shade stress [15]. Diploid *Populus ussuriensis* Kom. suffered more severe salt injury than polyploid [16]. What is more, tetraploid plants could be used as breeding material and bred with diploid ones to produce triploid plants [17], while triploids have the characteristics of few seeds [18]. For the fruit utilization of *P. edulis*, the seed is an important influence factor; thus, the study of tetraploid *P. edulis* helps establish the triploid breeding system and reduces the cost and difficulty of processing. Based on the importance of using polyploidy technology in promoting plant breeding, scientists are trying to achieve the goal of improving target plants quickly through artificial induction methods [19,20]. Colchicine induction is one of the most commonly used polyploid induction methods. It is applied to a wide range of crops, including *Actinidia chinensis* [21], *Cerasus humilis* [22], and Chinese jujube (*Ziziphus jujuba* Mill.) [23].

In nature, the wild polyploid *P. edulis* (2n = 18) is still absent. Zhangqin once used three kinds of passionfruit (*P. edulis*, *P. edulis* var. *flavicarpa* Deg., and Tai-Nung No. 1) to induce artificial triploids and tetraploids by culturing the endosperm of mature seeds and treating the adventitious buds with colchicine, respectively. She conducted a preliminary morphologic evaluation of the polyploidies, and the results showed that noticeable differences in leaf morphology and stoma were observed in the polyploid passionfruit, compared to the diploid [24]. However, the emphasis of Zhang’s research was more on the in vitro induction of polyploid passionfruit, and there is a lack of specific and further analysis in terms of the physiological and biochemical traits of tetraploid *P. edulis*. In recent years, the differences in morphology, chlorophyll content, and chlorophyll fluorescence parameters and the photosynthetic capacity variation rules of triploid *P. edulis* “Mantianxing” seedling leaves were revealed; as these findings demonstrated, the triploid plant was different from the diploid one in terms of its phenotype and superior to the diploid one in physiological indexes [25]. However, it is unknown whether tetraploid *P. edulis*, like most plants, has a decided polyploidization advantage in morphology, physiology, and biochemistry compared with its diploid counterpart. Thus, this research used tetraploid *P. edulis* seedlings as materials, conducted a relatively complete study on the above indexes, and further evaluated the performance of diploids and tetraploids under cold stress. It makes up for the gaps in related fields to a certain extent.

Temperature is one of the most vital factors which regulate plant development. Low temperatures can cause severe and irreversible damage to plant cells and induce physiological and metabolic changes affecting plant growth [26]. As a tropical and subtropical plant [27], low temperature is the main barrier for *P. edulis* to expand its habitat. Thus, to explore the adaptability of *P. edulis* to low temperatures, we analyzed the performance of photosynthetic characteristics and the activity of antioxidant enzymes, Malondialdehyde, and proline in diploid and tetraploid *P. edulis* under cold stress. This research aims to provide insights into the variations of *P. edulis* with different ploidies and offer a reference for expanding the variety of breeding in cold areas.

## 2. Results

### 2.1. Ploidy Levels Identification

Flow cytometry and chromosome counting were used to verify the ploidy levels. The results showed that, in flow cytometry, the fluorescence intensity in diploid leaves was 3665.0 while that in leaves treated with colchicine was 6920.5, presenting a nearly doubled 2C-DNA compared with the former. The diploid plants contained 18 chromosomes, while the variant plants had 36 chromosomes. These results suggested that *P. edulis* plants were tetraploid plants after being treated with colchicine (Figure 1).

From Figure 2, it can be observed that the tetraploids indeed had larger but fewer stomas than the diploid. The means of stoma length and width of the tetraploids significantly (*p* < 0.05) increased by 44.78% and 65.72%, respectively, as compared to those of the diploid plants. In contrast, the diploid had higher stoma numbers and density (Figure 2, Appendix A).

### 2.2. Morphological Characteristics Comparison of Diploid and Tetraploid P. edulis

Primary observation found significant morphological differences between the diploid and tetraploid plants (Figure 3). Overall, compared with the diploid, the tetraploid plants had the dwarfing phenotype, thicker ground diameter, darker leaf color, more leaf curves, and increased root length but fewer root numbers. Results indicated that polyploidization led to plant dwarfing but facilitated the development of the root system. Additionally, the increase in genome DNA content significantly influenced the leaf morphology. Moreover, the ground diameter of the tetraploids was 23.58% thicker on average than that of the diploid, and the highest value of ground diameter was tetraploid T3 (Appendix A).

#### 2.2.1. Root System Comparison of Diploid and Tetraploids

The differences between the underground parts of different ploidy levels were observed. The measurement of the root system showed that the means of the three tetraploid lines were 31.39% longer than that of the diploid, with T1 presenting a significant (*p* < 0.05) difference. However, the average root numbers for the diploid plant significantly (*p* < 0.05) increased by 183.64%, compared to the tetraploids. In a word, tetraploids had longer but fewer roots (Appendix A).

#### 2.2.2. Leaf Morphology

Leaf morphology comparison with more details suggested that the tetraploid plants had significant differences with the diploid in leaf thickness, leaf width, petiole length, leaf area, leaf serration numbers, and leaf color. Compared with the diploid, the means of leaf thickness, leaf width, petiole length, and leaf area in the tetraploids were increased by 19.22%, 18.90%, 51.22%, and 14.72% more than the diploid, respectively. There was no significant difference in leaf length between the two ploidy levels. With each ploidy level, the leaf color tends to be darker; tetraploid plants are all dark green, while diploid plants are dark yellow-green (Appendix A and Figure 4).

#### 2.2.3. Variation Analysis of Diploid and Tetraploid Morphological Indices

The variation analysis of the morphological indices of diploid and tetraploid *P. edulis* (Table 1) indicated that the most significant coefficient of variation was RL and RN in the diploid and tetraploid plants, respectively. In contrast, the smallest coefficient of variation was LA. However, except GD, RL, and PL, the coefficient of variation of RN, PH, LA, LW, IN, LSN, LL, and LT in tetraploid was higher than that of diploid, respectively.

### 2.3. Variation of Physiological Indices

#### 2.3.1. Soluble Sugar and Soluble Protein Contents

We examined the changes in soluble sugar and soluble protein content between the diploid and tetraploid plants. The mean soluble sugar contents in tetraploid T1, T2, and T3 were significantly (*p* < 0.05) raised by 65.01%, 62.79%, and 63.60%, respectively, as compared to the diploid. Results also showed a similar trend in the changes in soluble protein content among the two ploidy levels; for example, the soluble protein content in the diploid measured as 35.33 ± 4.87 mg·g^−1^, while the soluble protein contents in tetraploid T1, T2, and T3 were 30.46%, 27.69%, and 33.75% higher than the diploid, respectively (Table 2). These results indicated that the soluble sugar and protein content showed an absolute advantage after *P. edulis* polyploidization.

#### 2.3.2. Variation Analysis of Chlorophyll Content

Genome multiplication also affects chlorophyll content, which is essential for the promotion of plant growth and development. Compared with diploid, tetraploid significantly (*p* < 0.05) had more abundant chlorophyll content. The mean content of chlorophyll *a* in the tetraploids increased by 20.12% over the diploid. The mean content of chlorophyll *b* increased by 21.53% over the diploid. The mean of total chlorophyll content in the tetraploids was 20.48% higher than in the diploid. The average carotenoid content of the tetraploid lines was 28.97% higher than that of the diploid lines. These results can be summarized by saying that the tetraploids had richer chlorophyll *a*, chlorophyll *b*, total chlorophyll, and carotenoid content than the diploid (Table 2). This indicates that tetraploid is equipped with a better light capture capacity.

### 2.4. Difference in Antioxidant Enzyme and MDA, Pro Content in Diploid and Tetraploid

We tested the contents of superoxide dismutase (SOD), peroxidase (POD), and catalase (CAT) in the two ploidies. The average SOD content of tetraploid was 21.45% higher than that of diploid. The activity of POD had significantly (*p* < 0.05) increased after tetraploidization. Its average content in the three tetraploid lines was 50.49% higher than in the diploid. The mean CAT content of the tetraploids was 26.73% higher than that of the diploid, especially the T3 line, which had a conspicuous increase in advantage; the increased amplitude was 31.81% higher than that of the diploid (Figure 5). In general, the activity of the antioxidant enzyme of tetraploid was superior to that of diploid, and we can make a forecast that tetraploids of *P. edulis* may have a better performance than diploids under abiotic stress, according to the critical function of these enzymes in plant growth and development.

To reveal the stress resistance of the two ploidy levels of *P. edulis*, we tested the content of MDA and Pro. The results indicated that the diploid had a higher MDA content than all three tetraploids, which was 14.99% higher than the mean of the tetraploids, indicating that the damage degree of the diploid was more severe than the tetraploid in some stress conditions. The change trend in Pro content was different from MDA, compared with diploid; the tetraploids’ average Pro content was 44.56% higher than the diploid, which can be used to adjust the stability of the membrane (Figure 5). These results indicated that tetraploid *P. edulis* may have better plant stress resistance.

### 2.5. Comparison of Photosynthetic Characteristics and Chlorophyll Fluorescence Parameters between Diploid and Tetraploid P. edulis

#### 2.5.1. Comparison of Photosynthetic Characteristics

According to Figure 6, the trends in diurnal variation of *P*_n_, daily variation of *T*_r_, *G*_s_, and daily variation of *C*_i_ in diploid and tetraploid *P. edulis* were similar. As went on, the trends in *P*_n_, *T*_r_, and *G*_s_ in both the diploid and tetraploids presented as bimodal curves, and all increased to the peak at 10:00, and to the valley at 14:00, when the average *P*_n_ of tetraploid was 43.62%, and 83.14% higher than that of diploid, respectively (Appendix A). Overall, the *P_n_* of the three tetraploids was significantly (*p* < 0.05) higher than the diploid at each time, which means an increase in ploidy level may cause the improvement of the net photosynthetic rate in the plant. At 10:00 and 16:00, the average *T*_r_ of tetraploid was 54.82% and 41.71% higher than that of diploid, and the average *G_s_* of tetraploid increased by 13.25% and 16.10%, respectively (Appendix A). The *T*_r_ of the three tetraploids was significantly (*p* < 0.05) higher than the diploid at each time, except at 18:00, which may relate to the size of the plant’s stoma. The daily variation trend of *C*_i_ in diploid and tetraploid were very different from the other three trends, shown as capital letter “U”. From 8:00 to 14:00, the value of *C*_i_ in the two ploidy species declined continually and reached the minimum value at 14:00. From 14:00 to 18:00, the concentration of *C*_i_ in both diploid and tetraploid increased continually. On the whole trend, the *C*_i_ values of the tetraploids were all significantly (*p* < 0.05) greater than that of the diploid (Appendix A).

#### 2.5.2. Comparison of Chlorophyll Fluorescence Parameters between Diploid and Tetraploid *P. edulis*

As shown in Table 3, a significant difference existed in the initial photochemical efficiency and initial fluorescence parameters between diploid and tetraploid. Compared with diploid, the average value of *F*_o_ in tetraploid was 27.85% higher than diploid, especially the T1 line, which was significantly (*p* < 0.05) higher than the latter. Similar results were observed in other parameters, the average values of *F*_m_, *F*_v_, *F*_m_/*F*_o_, *F*_v_/*F*_m_, and *F*_v_/*F*_o_ in tetraploid were 17.40%, 19.10%, 46.62%, 16.63%, and 14.96% higher than those of diploid, respectively.

#### 2.5.3. Kinetic Curve of Chlorophyll Fluorescence Induction

Green plants contain many chloroplast stromata, which can produce red and changing fluorescence under light. With time exposure, the fluorescence changes and is presented as a dynamic curve, called the kinetic curve of chlorophyll fluorescence induction (OJIP curve). As Figure 7 shows, as time went on, the fluorescence intensity of diploid and tetraploid *P. edulis* increased continually and presented differences at the “J” point. During the whole process, the fluorescence intensity of the tetraploid was higher than that of the diploid, especially at the “I” and “P” points. Based on this analysis, we could conclude that tetraploid *P. edulis* had a larger fluorescence yield and better photosynthetic efficiency than diploid.

### 2.6. Correlation Analysis and Principal Component Analysis of Various Indices between Tetraploid and Diploid P. edulis

Significant differences were found between diploid and tetraploid when their morphological and physiological indices were compared. To explore the correlations and rules in these indices of diploid and tetraploid, correlation analysis was performed on the 28 indexes of the diploid and tetraploid of *P. edulis*. The results showed that plant height and internode numbers were significantly positively correlated with stoma number and density; on the contrary, they were negatively correlated with stoma length and width, chlorophyll content, soluble sugar, soluble protein, and photosynthetic characteristics. Leaf area positively correlated with chlorophyll content. Stoma size was very significantly positively correlated with chlorophyll content and carotenoids. This suggested that the larger the stoma size, the higher the chlorophyll content, and the better the photosynthetic characteristics. Stoma number and stoma density were very significantly negatively correlated with chlorophyll *a*, chlorophyll *b*, total chlorophyll content, and carotenoids. Chlorophyll *a*, chlorophyll *b*, total chlorophyll content, and carotenoid content were positively correlated with soluble sugar, soluble protein, *P_n_*, *T*_r_, *G_s_*, and *C*_i_. The content of chlorophyll *a* was positively correlated with *F*_m_ and *F*_v_. An extremely significant positive correlation existed between the total chlorophyll content, carotenoids, and *F*_o_, *F*_m_, and *F*_v_. The results of soluble sugar and soluble protein analysis showed that soluble sugar content was positively correlated with *P_n_*, *T_r_*, *G_s_*, *C_i_*, and *F*_o_. The same correlation was observed in the soluble protein and *P_n_*, *T_r_*, *C_i_*, *F*_o_, and *F*_v_ (Figure 8). To further analyze the relationship between each index, PCA (principal component analysis) was carried out (Appendix A). According to the result, there were seven components with eigenvalues greater than one, and the cumulative contribution rate reached 77.98%, which contained almost the information of the original data and could be used to evaluate and judge the germplasm resources of *P. edulis*. The first principal component was highly correlated with chlorophyll content, stoma width, carotenoid, and stoma length. These traits were mainly involved in the process of photosynthesis and respiration. The second principal component was highly related to leaf area, length, width, and stoma density. From Appendix A, there was a clear separation between diploid and tetraploid T1, T2, and T3 in the coordinate system, and three tetraploid lines showed higher overlap ratio, indicating a significant difference between diploid and tetraploid, and no significant difference between tetraploid lines.

### 2.7. The Analysis of Diploid and Tetraploid P. edulis under Cold Stress

#### 2.7.1. The Photosynthetic Characteristics Variation of Different Ploidy *P. edulis* under Cold Stress

Figure 9 shows that cold stress specifically affected diploid and tetraploid *P. edulis*. With the stress time prolonged, the damage degree increased, and the net photosynthetic rate, stoma conductance, and transpiration rate in both diploid and tetraploid leaves showed a downward trend and hit its lowest point at 48 h. During the whole stress treatment, the net photosynthetic rate in both diploid and tetraploid rapidly decreased in the 0~24 h, then slowing the decline, and all the *P_n_* of the three tetraploid lines were significantly (*p* < 0.05) higher than the diploid. However, there was no significance between tetraploid plants. At 48 h, the values of *P_n_* in T1, T2, and T3 were almost three times those of diploid. The trend in variation of *T*_r_ was similar to *P_n_*; at 48 h, the values of *T*_r_ in tetraploid T1, T2, and T3 were 125.35%, 105.63%, and 114.08% higher than diploid, respectively. Stoma is the channel through which water vapor and CO_2_ exchange between plants and the atmosphere, which also affects the photosynthesis and transpiration of plants. From Figure 9, the values of *G_s_* in diploid and tetraploid decreased with the stress times. However, there was significant variation in both ploidies, indicating that the stress effect of the photosynthetic efficiency under low temperatures was primarily related to stoma inhibition. Unlike the former three indices, the *C*_i_ of the two ploidy plants increased with the stress time; the *C*_i_ of diploid and tetraploid T1 lines rose significantly in the early stage of treatment but not significantly in the late stage. The intercellular CO_2_ concentration of tetraploid T2 and T3 lines increased significantly each time.

#### 2.7.2. The Changes in Enzyme Activity (SOD, POD, and Pro) and MDA Content of Diploid and Tetraploid *P. edulis* under Low-Temperature Stress

The variations of enzyme activity in the diploid and tetraploid plants are shown in Figure 10. The enhancement of SOD activity can improve oxidation resistance and stabilize membrane permeability in plants. Compared to the diploid, the SOD activities of the tetraploid plants were significantly (*p* < 0.05) higher. The SOD activity in the T1, T2, and T3 plants increased by 23.88%, 22.08%, and 21.07% more than the diploid at 48 h, respectively. The POD activity in the diploid and tetraploids rose rapidly in the first 12 h, and tetraploid plants maintained a significantly (*p* < 0.05) higher activity than the diploid from the 6 h to the end. At 48 h, the POD activity of tetraploid T1, T2, and T3 plants was 18.67%, 17.94%, and 18.12% higher than the diploid plant, respectively. That means the tetraploids performed a better active oxygen scavenging capacity under stress. The changing trend in the activity of CAT was similar to POD; the CAT activity of the tetraploids T1, T2, and T3 was 24.66%, 25.00%, and 24.57% higher than the diploid at 48 h, respectively. From these results, it can be concluded that tetraploids have more advantages in terms of antioxidant balance under cold stress (Figure 10). In general, plants with strong stress resistance will produce more Pro to reduce water potential and regulate the osmotic balance of plants. Pro activity is crucial for plants to improve cold resistance. Tetraploid plants showed a significantly (*p* < 0.05) higher Pro activity than diploid, especially during 12~48 h. At 48 h, the Pro activity of tetraploid T1, T2, and T3 plants were 67.65%, 68.59%, and 69.20% higher than diploid.

Malondialdehyde (MDA) is one of the commonly used indices to investigate the aging and resistance of plants. Cold stress increased MDA in the two ploidy levels, and there was no significant variation in the tetraploid plants during each stress period. In the diploid plant, the amount of MDA increased by 76.59% at 48 h compared to 0 h, and its increase amplitude was significantly higher than the tetraploid plants, indicating diploid plants are more vulnerable to cold stress than tetraploids (Figure 10).

## 3. Discussion

### 3.1. Morphological Characteristics Analysis

Multiple researchers have revealed that polyploidization changes plants’ morphology, physiology, and biochemistry [28,29] and also contributes to plant diversification and evolution [30]. For example, the tetraploid birch (*Betula pendula*) showed a significantly more extensive leaf area than its diploid counterpart [31]. According to previous studies, the so-called “gigas effect” may explain the more giant leaves in tetraploids; increased polyploid cell size could result in more giant leaf organs [32]. According to Zhang’s results, polyploid passionfruit (*P. edulis*, *P. edulis* var. *flavicarpa* Deg., and Tai-Nung No. 1) exhibited gigantism, with characteristics such as fewer and larger stomata, thicker and darker leaves, and a curved shape observed in tetraploids. Meanwhile, diploid plants presented flat and smooth leaves [24]. These results were consistent with the findings of this study. However, the difference between triploid and tetraploid *P. edulis* still needs further study. Moreover, we further analyzed the coefficient of variation of most of the morphological indexes (plant height, leaf area, root number) in tetraploid and diploid plants. The results verified tetraploids had more morphological variations than diploids (Table 1). Novel phenotypic variation may provide a selectional basis for crop improvement [33]. It is suggested that tetraploid plants can provide more plentiful choices for breeders to promote the breeding programs of new cultivars of *P. edulis*. It has been reported that tetraploid *Zizyphus jujuba* Mill. cv. *Zhanhua* had significantly larger but fewer stomas, and the grafted tetraploid plants onto mature trees of *Z. jujuba* Mill. cv. *Zhanhua* resulted in thicker stems, rounder, and succulent leaves compared to their diploid counterparts [34], suggesting tetraploid *P. edulis* could be used as a novel grafting stock material in the breeding program of *P. edulis*. This provides new research ideas for breeders.

Many mechanisms may cause plant phenotype changes. According to previous studies, DNA sequence, cis- and trans-acting effects, chromatin modifications, RNA-mediated pathways, and regulatory networks modulate the differential expression of homoeologous genes and phenotypic variation [35], as well as the phenotype variation related to the allele-dosage effects [36]. For example, the changes in (*fw2.2*) alleles influenced the fruit size in tomatoes [37]. Additionally, according to Wang’s results, differentially abundant proteins that are related to cell division were found to be more abundant in tetraploid *Paulownia australis* than in diploid. These proteins play a fundamental role in plant organ growth and development and could contribute to the variability in leaf traits. [38]. However, the regulation mechanism behind the phenotypic variation of tetraploid *P. edulis* still needs more in-depth study.

### 3.2. Physiological Traits

Multiplying the number of chromosomes in *P. edulis* affects physiological traits. Soluble sugar and protein are essential substances in promoting plant growth and development. Soluble sugar can provide enough nutrients, which further contributes to building macromolecules and energy for specific and coordinated development [39]. Soluble protein is related to osmotic regulation in plant cells and plays a critical role in plant abiotic stress response [40]. Supported by previous studies, the increase in soluble protein content was highly associated with the cold resistance of plants [41]. According to our results, the content of soluble sugar and protein in *P. edulis* increased with the increase in ploidy level. The average soluble sugar and soluble protein in tetraploid increased by 63.80% and 30.63%, respectively, compared with diploid (Table 2). It helps provide more energy for tetraploid *P. edulis*’ growth and development. Similar results also had been obtained in the tetraploid Fig tree (*Ficus carica* L.); its total soluble sugar content increased by 17.5% and 22.8% in 4× plants of “Sabz” and “Torsh”, respectively, compared with their original diploid counterparts. Furthermore, both tetraploid lines accumulated a higher total soluble protein [42]. Under drought stress, higher soluble sugar, protein, and proline content were monitored in autotetraploid *Ziziphus jujuba* Mill. var. *spinosawere* compared to its diploid counterpart, suggesting the former had better drought resistance [43].

After chromosomes doubled, changes in pigment content in the plant leaves were observed, which may change the color of plant leaves and affect the photosynthesis of plants. Multiple researchers found that increased chlorophyll content was measured in polyploid plants, which resulted in darker green leaves [44,45], suggesting darker leaves are a common characteristic of tetraploids. The chlorophyll content is directly related to chloroplast numbers; the latter could be used as the identification index of tetraploid [46] and plays a vital role in the photosynthesis system [47]. Our results showed that the chlorophyll *a* and chlorophyll *b*, total chlorophyll content, and carotenoid content in tetraploid were significantly higher in triploid leaves compared to diploid leaves with varying degrees of increase (Table 2), indicating that the tetraploid *P. edulis* possessed a stronger light absorption capacity and photosynthetic capacity than the diploid *P. edulis*. Our findings were consistent with a report on tetraploid *Lilium* FO hybrids [48]. Moreover, the *F*_v_/*F*_m_ and *F*_v_/*F*_o_ values in tetraploid were significantly higher than diploid, suggesting that tetraploid had a higher light conversion efficiency. In terms of physiological characteristics, both our result and Wang’s study referred to polyploid plants as having better physiology parameters. However, Wang’s analysis focused on the monthly changes in triploid and diploid “Mantianxing” [25].

### 3.3. Analysis of Different Ploidy Antioxidant Enzymes, MDA, and Pro Contents

The activity of antioxidant enzymes, malondialdehyde, and proline can reflect the sensitivity of plants to environmental factors and are often used as indicators to judge the strength of plant self-regulation. Plants usually remove excess ROS by enhancing their antioxidant enzyme activity to maintain cellular homeostasis and reduce the damage caused by stress [49,50]. In this study, the SOD, POD, and CAT activities in tetraploid *P. edulis* lines significantly increased compared with those in diploid, indicating that tetraploid *P. edulis* had a more robust capacity for removing free radicals and restoring equilibrium. This finding was consistent with the responses of the enzyme activity in tetraploid *Brassica. Juncea* under salinity stress [51]. Proline can protect and stabilize reactive oxygen species scavenging enzymes, and stabilize protein and cell structure, thereby maintaining cell osmotic potential. MDA is widely used to indicate the degree of injury caused by abiotic stress [52]. In this study, the content of MDA in tetraploid was lower than in diploid, while the former had higher Pro content (Figure 5). Therefore, we proposed that tetraploid *P. edulis* had lower damage and stronger adjustment ability to adverse environments than diploids.

### 3.4. Photosynthetic Characteristics and Chlorophyll Fluorescence between Diploid and Tetraploid P. edulis

The stoma and leaf play critical roles in photosynthesis. The stoma controls the gaseous exchange between the leaf and the external atmosphere [53]. The leaf is the principal place of photosynthesis in plants. In our study, tetraploids had a significantly longer and broader stoma and larger leaf area than the diploids. In addition, their chlorophyll and carotenoid content were significantly higher, contributing to plants’ photosynthesis. These findings indicated that the photosynthetic characteristics of tetraploid *P. edulis* were significantly better than diploid. Similar findings have been reported in tetraploid *Anoectochilus roxburghii* [54] and tetraploid *Liriodendron sino-americanum* [55].

In the diurnal variation of photosynthesis of most plants, the variation of photosynthetic rate is a “single peak” or “double peak” curve with a midday depression feature. This can be seen, for example, in the reports of *Camellia sinensis* (L.) O. Kuntze [56] and marula *Sclerocarya birrea* (A. Rich.) [57]. In this study, the diurnal variation of net photosynthetic rate, transpiration rate, and stomatal conduction degree day of diploid and tetraploid plants of *P. edulis* presented a “double peak” curve, and the diurnal variation of intercellular CO_2_ concentration presented a “U-shaped” curve. The lowest value was reached at 14:00 (Appendix A). This is consistent with the daily variation trend in plant photosynthesis studied by predecessors.

In the current study, the PS II primary photochemical efficiency and initial fluorescence parameters in the tetraploid lines were significantly higher than in the diploid line. The energy distribution ratio in the PS II reaction center and “OJIP” curve of the tetraploid lines were better than those of the diploid line (Figure 7), indicating that the tetraploid line had higher photochemical efficiency, fluorescence yield, and photosynthetic efficiency. The fluorescence parameters of triploid *P. edulis* also showed polyploid superiority [25]. Compared with diploid, the increase in chromosomes in *Plumbago auriculata* Lam. changed the physiology characteristics, and tetraploid plants showed better PSII activity and cold resistance [58].

### 3.5. Effects of Low-Temperature Stress on P. edulis

Temperature plays a critical role in the photosynthesis of plants [59]. Most plants can adjust their photosynthetic characteristics to acclimate their growth temperatures (temperature acclimation) [60]. Distinct plant species with varying ploidies exhibit differing regulatory capacities under abiotic stress conditions. Thus, the cold resistance evaluation of tetraploid *P. edulis* enriches the understanding of its physiological and biochemical adaptation to low temperatures, and provides insight for breeders to expand the range of *P. edulis* cultivation. Our study revealed that, under low-temperature stress, tetraploid *P. edulis* showed a superior *P_n_* (Figure 9), indicating stronger photosynthetic capacity, higher intrinsic photochemical activity, enhanced electron transfer efficiency, and photochemical efficiency, all of which contribute to the net photosynthetic rate. Our findings were supported by similar results obtained in low-temperature experiments. For example, triploid citrus varieties appeared more tolerant to naturally low temperatures than diploid ones, as evidenced by better photosynthetic properties (*P*_net_, *G*_s_, *F*_v_/*F*_m_, ETR/*P*_net_ ratio) [61]. Furthermore, *Dianthus broteri* had particular photochemical adaptations in higher ploidy levels under climate change [62].

Antioxidant enzyme variations in plants are considered scientific indicators for their response to cold stress. In the evaluations of the abiotic stress resistance of polyploid plants, it can be found that most plants with high ploidy often have higher antioxidant enzyme activity to maintain better resistance than ones with low ploidy. For example, autotetraploid *Poncirus trifoliata* enhanced its drought resistance by maintaining ROS detoxification and osmotic adjustment via elevating antioxidant activity and sugar accumulation compared its diploid counterpart [63]. Our findings demonstrated that tetraploid *P. edulis* was more adaptable to low-temperature stress.

Plant polyploidization is an essential force in plant evolution. Studies about plant polyploidization help biologists understand how plants have evolved and diversified and assist breeders in designing new strategies for crop improvement [64]. Thus, polyploid *P. edulis* with excellent properties can be used as experimental material to produce new polyploid *P. edulis* varieties.

## 4. Materials and Methods

### 4.1. Plant Materials and Growth Condition

The sterile seedlings (2n = 2x = 18) were obtained from the mature seeds of *P. edulis* “Tai-Nung No. 1”, provided by the Institute of Tropical Crop Science of Yunnan Province. Tai-Nung No. 1 *P. edulis* is a new hybrid variety (purple fruit species (female parent) × yellow fruit species (male parent)) bred by the Fengshan Tropical Horticultural Research Institute in Taiwan in 1981 [65].

Polyploid plants were obtained by treating the indefinite buds of *P. edulis* sterile seedlings with colchicine in the previous study of our research group [66]. Flow cytometric analysis further confirmed the ploidy levels of the sterile seedlings and plants treated with colchicine. After determination, the diploid sterile seedlings served as contrast, and three tetraploid plants (2n = 4x = 36) were named T1, T2, and T3 plant lines, respectively. Then, all the seedlings were subcultured in MS medium (MS + 1.0 mg/L 6 − BA + 0.2 mg/L IAA, Sucrose 30 g/L, Agar 3.75 g/L, pH = 5.8~6.0) by using the indefinite bud proliferation method [66]. All experimental seedlings were cultivated in the tissue culture room of Southwest Forestry University with a temperature of (25 ± 2) °C and an average of 12 h/d of light/darkness, and the light intensity was 1000~1500 Lux.

### 4.2. Ploidy Verification of Tetraploid P. edulis

#### 4.2.1. Flow Cytometric Analysis

Two young leaves harvested from the top of each plant, including *P. edulis* sterile seedlings and variant seedlings treated with colchicine, were sent to Kunming Medical University for flow cytometry detection using the CyFlow Ploidy Analyzer (Sysmex Parec, Görlitz, Germany). The DNA content of *P. edulis* sterile seedlings was used as a contrast to detect the changes in the DNA content of the variant plants.

#### 4.2.2. Chromosome Counting

Chromosome counting was conducted to identify the chromosome numbers of the potential plants. From 10:00 a.m. to 11:00 a.m., 1–2 cm root tips of the diploid plants and plants treated with colchicine were collected and soaked in a mixed solution, which was made from the equivalent volume of 0.1% colchicine solution and 0.002 mol·L^−1^ 8-hydroxyquinoline to fix for three h. The pre-treated root tips were washed with distilled water 2~3 times, then fixed in Carnoy Fluid for 10 min and then rinsed with 90% and 70% alcohol successively for later use. The fixed root tips were cleaned with distilled water 2~3 times, followed by 1 mol·L^−1^ hydrochloric acid and a constant temperature water bath at 60 °C for 30 min to dissociate the root tips. The root crown from the root tip was removed, and 1–2 mm of the root tip was taken and placed on a microscope slide, mashed with tweezers, and stained in 1 drop of modified phenol fuchsin-staining solution, then we covered the slid with coverslip and gently tapped the coverslip with a pencil eraser to make the cells evenly dispersed and to observe the chromosomes.

### 4.3. Morphological Characterizations Measurement of Polyploidy P. edulis

The same batch of 45-day-old subculture seedlings with different ploidy was randomly selected for the comparison of morphological characteristics regarding plant height, ground diameter, leaf thickness, internodes, and root numbers. The Plant Image Analyzer (WanShen LA-S series; Hangzhou Wseen testing Technology Co., LTD, Hangzhou, China, http://www.wseen.com/, accessed on 27 August 2024) was used to scan the leaf samples and record leaf length, leaf width, petiole length, leaf area, and leaf serration number. Each plant was considered an independent biological replicate. We set three biological replicates.

### 4.4. Measurement of Physiological and Biochemical Attributes

Three full-expanded leaves with different leaf positions were collected from the same plant and thoroughly mixed. Then, they were divided into 3 independent biological replicates (weighted 0.1 g of fresh material per replicate total). All samples were weighed to determine the physiological and biochemical attributes of *P. edulis*.

The contents of chlorophyll *a*, chlorophyll *b*, and carotenoids have been determined by vis spectrophotometer (7230G, Shanghai Jinghua Technology Instrument Co., LTD, Shanghai, China, http://www.jinghuatec.cn/, accessed on 27 August 2024) according to Arnon [61]. The absorbance was determined at 663 nm, 645 nm, and 470 nm, respectively.

Following the manufacturer’s guidelines, we used corresponding commercial kits bought from Suzhou Michy Biomedical Technology Co., Ltd., (Suzhou, China, www.michybio.com, accessed on 27 August 2024) to measure the contents of soluble sugar (anthrone colorimetry), soluble protein (Coomassie Brilliant Blue G-250 method), malondialdehyde (MDA, thiobarbituric acid reactive substance assay), proline (Pro, sulfosalicylic acid (SA) method) content, catalase (CAT, Ammonium Molybdate Colorimetry method), superoxide dismutase (SOD, WST-8 method), and peroxidase (POD, Spectrophotometer, SpectraMax Plus 384, Moleculardevices (Shanghai) Co., LTD, China, https://www.moleculardevices.cn/, accessed on 27 August 2024). The specific operation steps were strictly implemented free of charge, following the instructions provided by the company. All the parameters were measured using the enzyme-linked immunosorbent assay.

### 4.5. Analysis of Photosynthetic Characteristics and Chlorophyll Fluorescence Parameters

The diurnal changes in photosynthetic characteristics of *P. edulis* were measured with the LI-6800 Portable Photosynthesis System (LI-COR, Lincoln, NE, USA), and the net photosynthetic rate (*P*_n_), transpiration rate (*T*_r_), stomatal conductance (*G_s_*), and intercellular carbon dioxide concentration (*C*_i_) of the leaves were mainly determined. Three seedlings in each plant line were collected, and three leaves with stable growth at the top of each seedling were used to determine the photosynthetic characteristics at 8:00, 10:00, 12:00, 14:00, 16:00, and 18:00, respectively. A plant efficiency analyzer (Handy PEA, Hansatech Instruments Ltd., Narborough Road, Pentney, King’s Lynn, Norfolk PE32 1JL, UK) was used to determine the chlorophyll fluorescence of diploid and tetraploid leaves. After 30 min dark adaptation, the leaves were measured using the instrument’s probe.

### 4.6. Comparison of Diploid and Tetraploid P. edulis under Low-Temperature Stress

The same batch of rooting seedlings with stable growth for 45 days was randomly selected and moved into a 4 °C refrigerator to implement cold stress, and the treatment time was set at 0 h, 6 h, 12 h, 24 h, 36 h, and 48 h, respectively. Photosynthetic characteristics and chlorophyll fluorescence parameters were determined immediately after cold stress treatment. Three leaves from the same plant were mixed to be divided into three independent biological replicates, and 0.1 g of material was weighed per replicate. All samples of different plant lines were loaded into labeled freezer tubes. These leaves samples were quickly cooled with liquid nitrogen and stored in a −80 °C refrigerator for further analysis. We determined the activity of SOD, POD, and CAT enzymes and the content of MDA and Pro by using the kit of Suzhou Michy Biomedical Technology Co., Ltd (Suzhou, China, www.michybio.com, accessed on 27 August 2024).

### 4.7. Data Processing and Analysis

Excel (2019) was used for data sorting and mapping, IBM SPSS statistics 22.0 was used for single-factor analysis of variance and two-factor variance analysis, the Duncan method was used for multiple comparisons, and Pearson correlation analysis was used to analyze the correlation. Origin (2021) and PowerPoint (2019) were used to draw diagrams. An independent samples *t*-test was used to analyze the mean value difference and test the significance of diploidy and tetraploidy. Each index’s calculation and analysis were conducted via average value, and each average value was the mean of three independent biological replicates.

## 5. Conclusions

To summarize, this study conducted a series of comparisons to investigate the differences in morphology, stoma, photosynthesis characteristics, chlorophyll content, chlorophyll fluorescence, antioxidant enzyme activity, and resistance in cold stress of tetraploid and diploid *P. edulis*. The results reveal that polyploidization not only leads to the dwarfing phenotype, bigger stoma with lower density, larger leaf with a darker color, and longer root, but also makes tetraploid *P. edulis* perform better in physiological and biochemical traits than its diploid counterpart. The improved soluble sugar and protein, chlorophyll content, and the active antioxidant enzyme are useful for tetraploids when it comes to developing an effective protective system under temperature stress. Our findings provide understanding for researching the differences between diploid and tetraploid *P. edulis* and a reference for breeding programs involving *P. edulis*.

## Figures and Tables

**Figure 1 plants-13-02603-f001:**
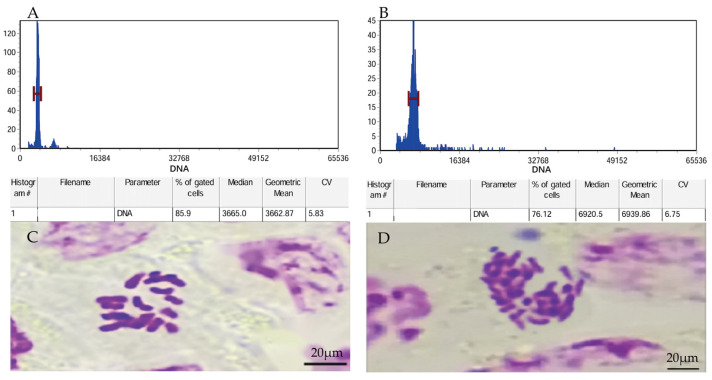
Flow cytometric profiles of diploid and *P. edulis* treated by colchicine (tetraploid). (**A**) The value corresponding to the diploid flow cytometer fluorescence peak on the abscissa. (**B**) The value corresponding to the tetraploid flow cytometer fluorescence peak on the abscissa. (**C**) Chromosomes prepared from diploid (2n = 2x = 18) and (**D**) chromosomes prepared from tetraploid (2n = 4x = 36) samples of *P. edulis*.

**Figure 2 plants-13-02603-f002:**
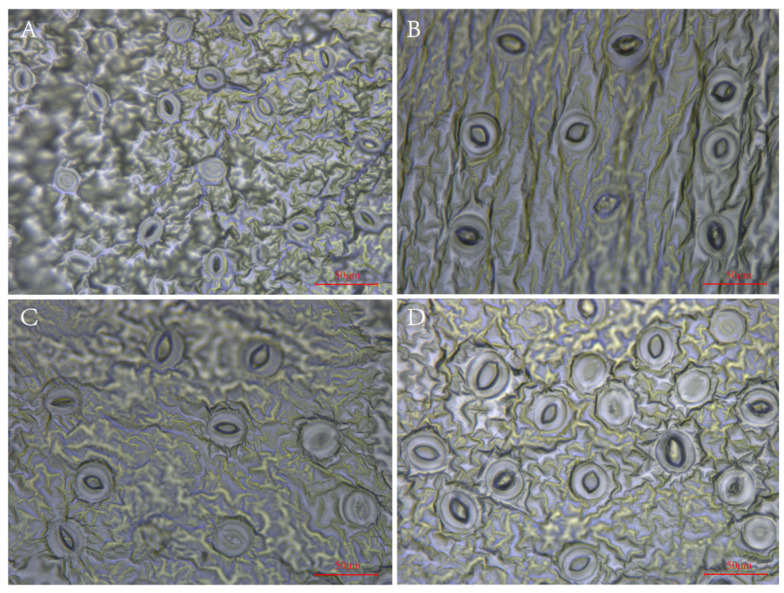
Stoma morphology of diploid and tetraploid *P. edulis*. (**A**) Stoma morphology of diploid. (**B**) Stoma morphology of tetraploid T1. (**C**) Stoma morphology of tetraploid T2. (**D**) Stoma morphology of tetraploid T3.

**Figure 3 plants-13-02603-f003:**
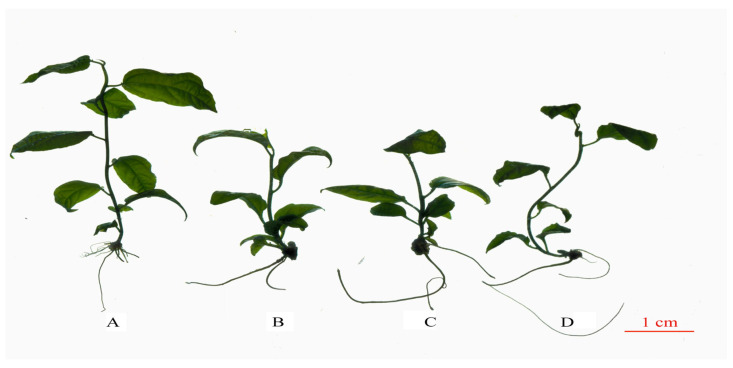
Morphological diagram of diploid and tetraploid *P. edulis*. (**A**) Morphological diagram of diploid. (**B**) Morphological diagram of tetraploid *P. edulis* T1. (**C**) Morphological diagram of tetraploid *P. edulis* T2. (**D**) Morphological diagram of tetraploid *P. edulis* T3.

**Figure 4 plants-13-02603-f004:**
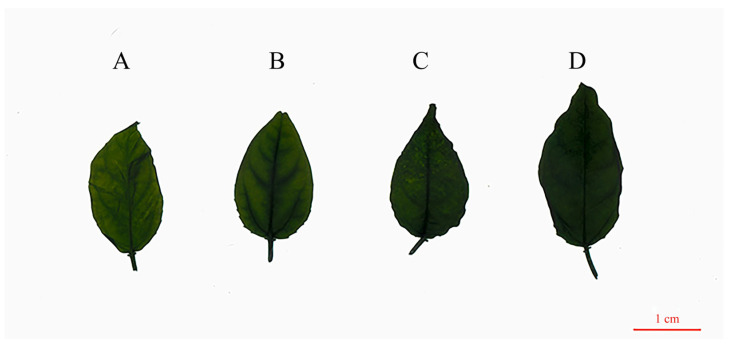
Leaf morphology of diploid and tetraploid plants of diploid and tetraploid *P. edulis.* (**A**) Diploid. (**B**) Tetraploid T1. (**C**) Tetraploid T2. (**D**) Tetraploid T3.

**Figure 5 plants-13-02603-f005:**
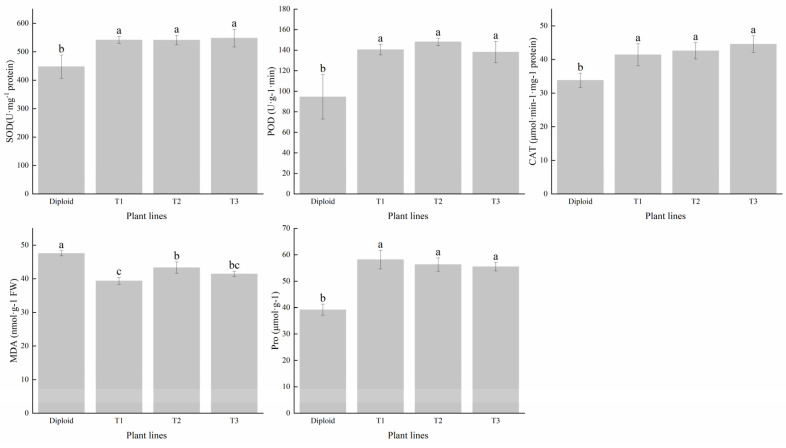
Difference analysis of antioxidant enzyme (SOD, POD, and CAT) and contents of MDA and Pro in diploid and tetraploid of *P. edulis*. Note: Different letters indicate significant differences, and the same letters mean insignificant differences (*p* < 0.05). Duncan test.

**Figure 6 plants-13-02603-f006:**
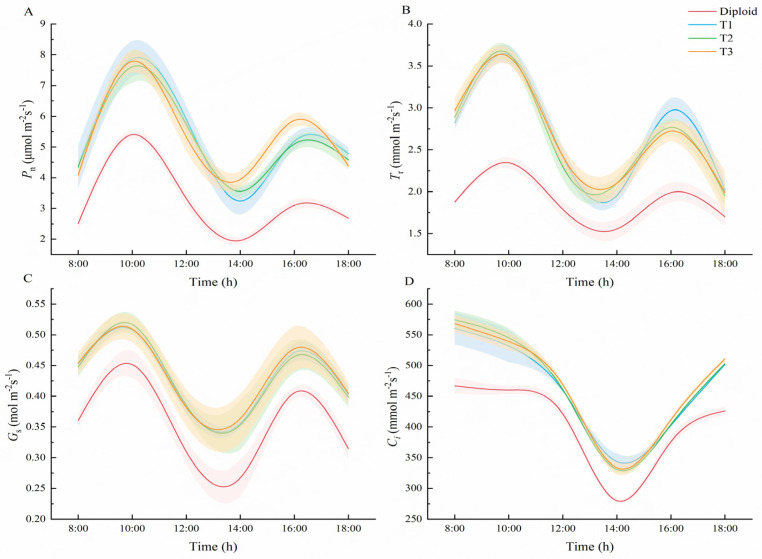
Comparison of photosynthetic characteristics between diploid and tetraploid. ((**A**) Diurnal variation trend in net photosynthetic rate of diploid and tetraploid *P. edulis*. (**B**) Diurnal variation trend in transpiration rate of diploid and tetraploid *P. edulis*. (**C**) Diurnal variation of stomatal conductance degree days of diploid and tetraploid *P. edulis*. (**D**) Diurnal variation trend in intercellular CO_2_ concentration in diploid and tetraploid *P. edulis.*)

**Figure 7 plants-13-02603-f007:**
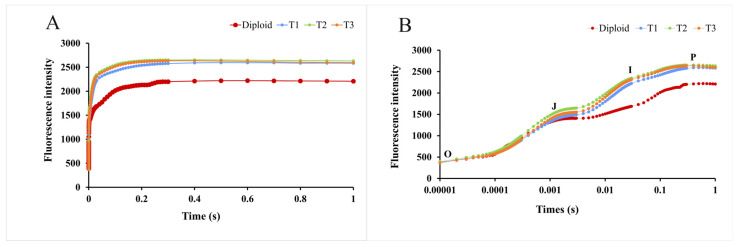
Kinetic curves of chlorophyll fluorescence induction of diploid and tetraploid *P. edulis*. (**A**) represents the difference in fluorescence intensity between diploid and tetraploid *P. edulis* on a linear time coordinate, and (**B**) represents the difference in fluorescence intensity between diploid and tetraploid *P. edulis* on a logarithmic time coordinate.

**Figure 8 plants-13-02603-f008:**
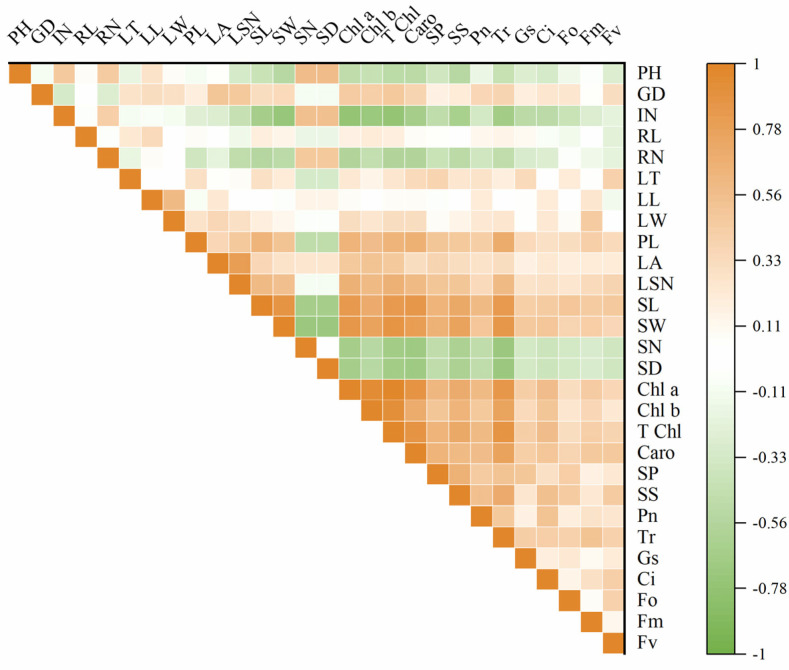
Correlogram displaying correlations among different studied characteristics of two *P. edulis* cultivars. Positive and negative correlations are represented in orange and green colors, respectively. The correlation coefficient based on the color index is shown on the right side of the correlogram. PH: plant height; GD: ground diameter; IN: internode number; RL: root length; RN: root number; LT: leaf thickness; LL: leaf length; LW: leaf width; PL: petiole length; LA: leaf area; LSN: leaf serration number; SL: stomata length; SW: stomata width; SN: stomata number; SD: stomata density; Chl *a*: chlorophyll *a*; Chl *b*: chlorophyll *b*; T Chl: total chlorophyll content; Caro: carotenoid, SP: soluble protein; SS: soluble sugar. *P*_n_: net photosynthetic rate; *T*_r_: transpiration rate; *G*_s_: stomatal conductance; *C*_i_: intercellular CO_2_ concentration; *F*_o_: minimum fluorescence yield in the absence of photosynthetic light; *F*_m_: maximum fluorescence yield in the absence of photosynthetic light; *F*_v_: variable fluorescence.

**Figure 9 plants-13-02603-f009:**
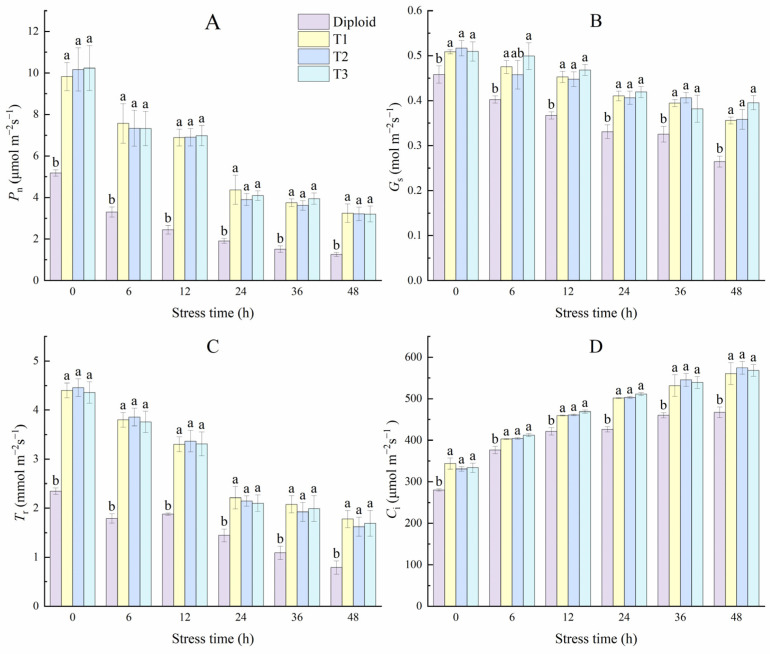
Effect of low-temperature stress on photosynthesis characteristics of diploid and tetraploid *P. edulis*. (**A**) Net photosynthetic rate (*P_n_*). (**B**) Stomatal conductance (*G_s_*). (**C**) Transpiration rate (*T_r_*). (**D**) Intercellular CO_2_ concentration (*C*_i_). Different letters indicate significant differences in the different plant lines at the same stress stage, and the same letters mean insignificant differences (*p* < 0.05).

**Figure 10 plants-13-02603-f010:**
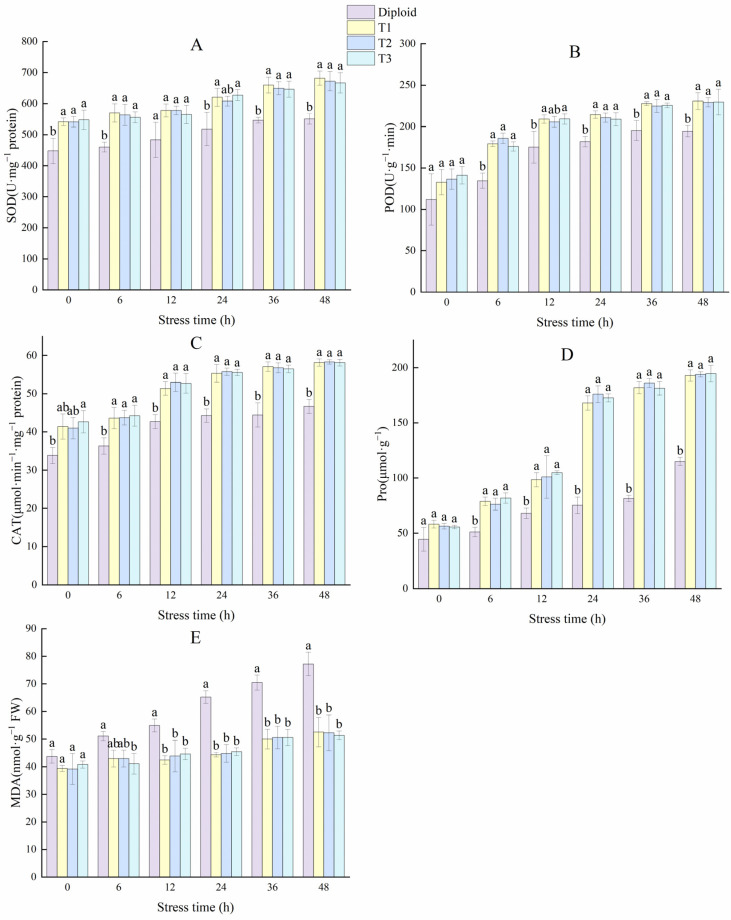
Effect of low-temperature stress on antioxidant enzymes (SOD, POD, CAT), Pro, and MDA content of diploid and tetraploids. (**A**) SOD (superoxide); (**B**) POD (peroxidase); (**C**) CAT (catalase); (**D**) Pro (proline); (**E**) MDA (malondialdehyde). Different letters indicate significant differences in the different plant lines at the same stress stage, and the same letters mean insignificant differences (*p* < 0.05).

**Table 1 plants-13-02603-t001:** Variation analysis of morphological indices of diploid and tetraploid *P. edulis*.

Index	Ploidy Level	Minimum Value	Maximum Value	Means ± Standard Error	Standard Deviation	Coefficient of Variation (%)
PH (cm)	2	6.12	11.28	8.47 ± 0.57 **	1.72	20.31
	4	2.46	10.27	5.81 ± 0.38	2.00	34.42
GD (cm)	2	0.94	1.75	1.27 ± 0.10	0.30	23.62
	4	1.08	2.11	1.56 ± 0.06 *	0.29	18.59
IN	2	5.00	8.00	6.22 ± 0.28 **	0.83	13.34
	4	3.00	6.00	4.15 ± 0.18	0.82	19.76
RL (cm)	2	1.21	10.83	4.66 ± 1.35	4.05	86.91
	4	1.11	18.32	6.12 ± 0.80	4.16	67.97
RN	2	2.00	13.00	5.78 ± 1.14 *	3.42	59.17
	4	0.00	7.00	2.04 ± 0.32	1.68	82.35
LT (mm)	2	0.07	0.14	0.10 ± 0.01	0.02	22.00
	4	0.07	0.21	0.12 ± 0.01 **	0.03	23.00
LL (cm)	2	2.14	3.32	2.84 ± 0.13	0.40	14.08
	4	1.92	3.89	2.89 ± 0.09	0.47	16.26
LW (cm)	2	1.12	1.82	1.39 ± 0.08	0.24	17.27
	4	1.23	3.62	1.65 ± 0.08	0.44	26.67
PL (mm)	2	3.55	5.99	4.58 ± 0.27	0.82	17.90
	4	4.37	9.13	6.92 ± 0.23 **	1.22	17.63
LA (mm^2^)	2	411.26	429.09	416.32 ± 2.03	6.10	1.47
	4	410.83	593.83	477.62 ± 13.94 **	72.44	15.17
LSN	2	3.00	5.00	3.89 ± 0.26	0.78	20.05
	4	4.00	10.00	6.59 ± 0.32 **	1.67	25.34

Note: PH: plant height; GD: ground diameter; IN: internode number; RL: root length; RN: root number; LT: leaf thickness; LL: leaf length; LW: leaf width; PL: petiole length; LA: leaf area; LSN: leaf serration number. * means the difference between diploid and tetraploid was significant (*p* < 0.05); ** means the difference was very significant (*p* < 0.01). Independent samples *t*-test.

**Table 2 plants-13-02603-t002:** Variation analysis of physiological index in diploid and tetraploid plants of *P. edulis*.

Ploidy Level	Line	Soluble Sugar Content (mg·g^−1^)	Soluble Protein Content (mg·g^−1^)	Chlorophyll *a* (mg·g^−1^)	Chlorophyll *b* (mg·g^−1^)	Total Chlorophyll (mg·g^−1^)	Carotenoids (mg·g^−1^)
2	Diploid	1.10 ± 0.05 b	35.33 ± 1.62 b	1.44 ± 0.04 c	0.51 ± 0.03 b	1.95 ± 0.04 c	0.40 ± 0.04 b
4	T1	1.81 ± 0.08 a	46.09 ± 1.52 a	1.65 ± 0.05 b	0.61 ± 0.03 a	2.26 ± 0.08 b	0.51 ± 0.04 ab
4	T2	1.78 ± 0.10 a	45.11 ± 2.08 a	1.69 ± 0.02 b	0.58 ± 0.02 ab	2.27 ± 0.01 b	0.55 ± 0.04 a
4	T3	1.80 ± 0.05 a	47.25 ± 1.82 a	1.85 ± 0.09 a	0.67 ± 0.03 a	2.51 ± 0.12 a	0.51 ± 0.03 ab

Note: Different letters indicate significant differences, and the same letters mean insignificant differences (*p* < 0.05), Duncan test.

**Table 3 plants-13-02603-t003:** Analysis of the difference in initial photochemical efficiency and initial fluorescence parameters between diploid and tetraploid PS II of *P. edulis*.

Ploidy Levels	Line	*F* _o_	*F* _m_	*F* _v_	*F*_m_/*F*_o_	*F*_v_/*F*_m_	*F*_v_/*F*_o_
2	Diploid	520.89 ± 29.88 b	1571.11 ± 58.70 b	1716.89 ± 71.49 b	2.35 ± 0.06 b	0.96 ± 0.05 b	2.67 ± 0.10 b
4	T1	693.00 ± 60.54 a	1802.78 ± 76.50 ab	2054.22 ± 115.85 a	3.29 ± 0.41 a	1.06 ± 0.03 ab	3.03 ± 0.20 ab
4	T2	641.11 ± 43.08 ab	1852.67 ± 88.95 a	2038.00 ± 72.60 a	3.36 ± 0.32 a	1.11 ± 0.04 ab	3.19 ± 0.15 a
4	T3	663.78 ± 52.05 ab	1878.22 ± 106.51 a	2042.22 ± 89.30 a	3.68 ± 0.33 a	1.17 ± 0.10 a	2.97 ± 0.20 ab

Note: *F*_o_, minimum fluorescence yield in the absence of photosynthetic light; *F*_m_, maximum fluorescence yield in the absence of photosynthetic light; *F*_v_, variable fluorescence; *F*_m_/*F*_o_, electron transfer rate; *F*_v_/*F*_m_, the maximum quantum yield of PSII; *F*_v_/*F*_o_, the ratio of photochemical to nonphotochemical processes. Different letters indicate significant differences, and the same letters mean insignificant differences (*p* < 0.05); Duncan test.

## Data Availability

All data generated or analyzed during this study are included in this article. All data are available from the first author upon request.

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
