# Peer review of "Identification and Evaluation of Diploid and Tetraploid Passiflora edulis Sims"

_plants, 2024, doi:10.3390/plants13182603_

Round 1

Reviewer 1 Report (Previous Reviewer 1)

Comments and Suggestions for Authors

The authors have addressed my concerns, and the paper is ready to publish

Author Response

Reviewer 2 Report (Previous Reviewer 2)

Comments and Suggestions for Authors

Just got around to this this morning. I am satisfied with the authors' response.

Comments on the Quality of English Language

I suggest having a copy-editor review this prior to publication.

Author Response

This manuscript is a resubmission of an earlier submission. The following is a list of the peer review reports and author responses from that submission.

Round 1

Reviewer 1 Report

Comments and Suggestions for Authors

Some minor suggestions made on the text, but generally well written.

It would be good to comment about fruit although the authors may not have taken plants to this stage.

Comments on the Quality of English Language

Generally good but a few instances where it can be improved

Author Response

Authors' responses attached.

Reviewer 2 Report

Comments and Suggestions for Authors

Review of “Identification and evaluation of diploid and tetraploid Passiflora edulis Sims” by Xin Su, et al. (Manuscript ID: plants-3060661)

OVERVIEW

The authors present a study where they induced polyploidy in three individuals of Passiflora edulis. They characterized several morphological, physiological, and chemical traits in vitro in the the tetraploids and one diploid for comparison, concluding that polyploid germplasm could be useful for future passionfruit breeding efforts. I think this is an interesting report that complements other studies of induced polyploidy and its effect on crop plants. However, the study itself is quite limited by a low replication overall and uneven replication among diploid and tetraploid accessions. Also, the methods are not reproducible as written, and the manuscript is difficult to follow at times. These points are discussed further below.

COMMENTS

Any conclusions of this study are extremely limited by its level of replication and cultural conditions. As I understand, the authors induced three tetraploid plants, then produced three clonal replicates from each of those plants via tissue culture (“the shoot proliferation method” was mentioned without elaboration on line 530) to use for trait measurements. Only a single diploid plant (represented by three clonal replicates) was used, so the authors were not able to report on the variation within the diploids and tetraploids equally, which comprises an inadequate control for any conclusions made about the tetraploids. The fact that all observations were made at the seedling stage under tissue culture conditions limits any claim about these varieties being useful for cultivar production, as measures could easily change in adult plants grown under field conditions where light, soil, humidity, and biotic conditions will be drastically different.

The materials and methods are lacking crucial information that would be necessary to repeat this experiment. No information about the germplasm (cultivar, original population, etc) aside from the donor source are included in the methods. I can’t find any information about the replication of plants used for polyploidy induction, flow cytometry, chromosome counting. Flow cytometry methods are limited to the instrument used and do not include the preparation protocol or fluorescent stain used. Neither an internal nor external standard was used for flow cytometry, and any inferred measures of DNA content or cytometric statistics (cell count, fluorescence CV, etc) are not reported (just mean fluorescence). The specific commercial kits used to quantify various compounds were not identified, only their supplier.

Comments on the Quality of English Language

The manuscript contains many grammatical errors and instances of unusual phrasing or word choice that decrease the manuscript’s readability. I advise using a copy-editing service to revise the manuscript prior to resubmission or publication

Author Response

Authors' responses attached.
